# Development and Validation of a Visualized Posture Risk Assessment Questionnaire for Low Back Pain in Daily Activities: A Study in Taiwan

**DOI:** 10.3390/healthcare12222274

**Published:** 2024-11-14

**Authors:** Yu-Tzu Chang, Yi-Ju Chen, Chinyu Ho, Chienyu Yeh, Cheng-Jung Huang, Jason Jiunshiou Lee

**Affiliations:** 1Department of Family Medicine, Taipei City Hospital Yangming Branch, Taipei 111, Taiwan; 2Department of Psychology, Soochow University, Taipei 111, Taiwan; 3Department of Health and Welfare, University of Taipei, Taipei 100, Taiwan; 4Department of Family Medicine, Lo-Sheng Sanatorium and Hospital, New Taipei City 242, Taiwan; 5Department of Physical Therapy, Taipei City Hospital Yangming Branch, Taipei 111, Taiwan; 6Institute of Public Health, National Yang Ming Chiao Tung University, Taipei 112, Taiwan; 7Department of Health Care Management, National Taipei University of Nursing and Health Sciences, Taipei 112, Taiwan

**Keywords:** low back pain, posture, daily activities, risk assessment, questionnaire validation, visualized questionnaire, Taiwan

## Abstract

Background/Objectives: A proper posture is essential for musculoskeletal health, and a poor posture can lead to low back pain. To address the limitations of traditional text-based questionnaires, this study developed and validated a visualized posture assessment questionnaire for evaluating daily postures related to low back pain. The questionnaire was administered in Taiwan and designed using Traditional Chinese language. Methods: The proposed questionnaire evaluates six categories of daily activities including lifting heavy objects, sitting, putting on shoes, face washing and tooth brushing, getting out of bed, and doing sit-ups, or similar actions. Each category comprises an ergonomic posture and a non-ergonomic posture with corresponding illustrations. The questionnaire was administered to 100 participants, and its internal consistency was evaluated using Cronbach’s α, while test–retest reliability was assessed using intraclass correlation coefficients (ICCs). An expert panel reviewed the content validity, and the item-level content validity index (I-CVI) was calculated for each item and illustration. Results: Testing revealed a Cronbach’s α of 0.808, indicating high internal consistency, and a test–retest reliability, as measured by ICCs, of 0.78, indicating high stability over time. The I-CVI scores were high across all items, with the illustrations unanimously rated by the experts as highly relevant, supporting the effectiveness of the questionnaire’s visualized format for enhanced comprehension. Conclusions: The proposed questionnaire exhibits high reliability and validity, rendering it effective in evaluating posture-related risks of low back pain. This questionnaire also offers a more accessible and intuitive alternative to text-based questionnaires, with potential applications in clinical and research settings.

## 1. Introduction

Maintaining a proper posture in daily life is crucial for musculoskeletal health [1,2]. A poor posture can lead to pain and musculoskeletal disorders, and low back pain is the most common disorder [3]. Low back pain is a common problem that affects individuals of all ages, and it poses a major challenge to healthcare and public health because of its effect on workforce productivity and the associated socioeconomic losses [4,5]. According to statistics, approximately 84% of adults experience low back pain at some point in their lives; moreover, low back pain is the second most common complaint in primary care [6,7]. Taiwan’s Ministry of Health and Welfare revealed that back diseases (e.g., disc disorders or back pain) accounted for 1.97% of all outpatient visits in 2019, ranking tenth among all diseases. Similarly, in 2015, 1.6% of patients in the United States sought medical care for back symptoms, which ranked tenth among all complaints [8]. A systematic review indicates that 12% of patients worldwide experience activity limitations due to low back pain lasting more than 1 day [9].

Postures involving sitting, standing, lying down, lifting heavy objects, bending, and twisting typically rely on back muscles. Inappropriate movements, such as excessive bending or twisting, can significantly increase compressive and shear forces on the lumbar spine. These forces strain the lumbar musculature, requiring the muscles to stabilize the spine under load. Over time, this increased demand on the lumbar muscles may lead to muscle fatigue, strain, and an elevated risk of low back pain. [10]. For example, frequent or incorrect bending or twisting motions can strain the lower back muscles, particularly when lifting heavy objects or standing on one leg wearing slippers or socks, which can excessively stretch the lumbar muscles [11]. In addition, slouching while sitting for long periods or assuming a forward head posture may be associated with low back pain. Similarly, standing with excessive lumbar lordosis (swayback) or assuming inappropriate postures while face washing or tooth brushing can contribute to back problems [12]. Sitting up directly after lying on the bed, instead of rolling to the side and using one’s arms to push up, can also often cause low back pain [13].

In addition to rehabilitation and medication, practical assessment tools are often used to identify the relationship between daily life postures and low back pain [14]. These tools play a key role in implementing proper posture education to support rehabilitation. Various tools are currently available for evaluating daily life postures including simple questionnaires, such as the Roland–Morris Disability Questionnaire (RMDQ) and the Nordic Musculoskeletal Questionnaire (NMQ), which assess musculoskeletal health and are widely used due to their validated reliability. Additionally, semi-visual tools, like the Rapid Upper Limb Assessment (RULA), provide a more detailed analysis of specific postures, particularly in occupational settings [15]. However, most of these tools focus on work-related or task-specific postures rather than daily life activities and may still rely heavily on text-based assessments, which can pose challenges for specific populations such as older adults or individuals with reading difficulties. [16,17]. While RMDQ and NMQ effectively assess pain and disability related to musculoskeletal disorders, they do not specifically address posture-related behaviors in daily activities.

Developing a daily life posture assessment questionnaire that integrates images to illustrate different postures and activities can considerably improve the respondents’ understanding and accuracy in answering the questionnaire. By integrating visual aids, this approach aims to increase accessibility and comprehension among populations such as older adults or individuals with reading difficulties, making the assessment of posture-related low back pain more inclusive. With such images, the respondents can easily identify and evaluate their postures, thereby mitigating the likelihood of misinterpretation and increasing the reliability of their responses [18]. A questionnaire that integrates images for assessing daily life postures is currently not available in Taiwan. Developing a questionnaire that combines images and text may thus offer a comprehensive approach for evaluating low back pain-related postures. Accordingly, this study developed a Traditional Chinese (Taiwanese) visualized posture assessment questionnaire and evaluated its reliability and validity in determining the relationship between daily life postures and low back pain.

## 2. Materials and Methods

### 2.1. Questionnaire Design

The proposed questionnaire is a self-made tool designed to evaluate daily life habits. Participants are asked to recall and examine the frequency of various actions or postures that they have performed or assumed over the preceding 2 weeks. This 2-week interval was chosen based on established recommendations in health research, where intervals of 1 to 2 weeks are commonly used [19]. The questionnaire comprises six categories of daily activities and postures: lifting heavy objects, sitting, putting on shoes, face washing and tooth brushing, getting out of bed, and doing sit-ups. Each category includes two indicators: an ergonomic posture and a non-ergonomic posture, with descriptions and illustrations for each. Ergonomic postures minimize strain on the musculoskeletal system, promoting safety and comfort. In contrast, non-ergonomic postures are associated with increased physical strain and a higher risk of discomfort or injury [20]. Each action posture is scored based on the difference in frequency between ergonomic and non-ergonomic postures in daily life. If the frequency of ergonomic postures is higher than that of non-ergonomic postures, a score of 2 points is assigned. If the frequency of ergonomic postures is lower than that of non-ergonomic postures, a score of 0 points is assigned. If the frequencies of ergonomic and non-ergonomic postures are the same, a score of 1 point is assigned. The total score of the questionnaire ranges from 0 to 12 points. The higher the total score, the less likely the actions and postures are to cause low back pain. The final questionnaire design was established after three rounds of discussion between the author, three rehabilitation specialist physicians, and physical therapists. The author drafted textual descriptions of the aforementioned postures, and professional graphic artists subsequently illustrated these descriptions. The illustrations were developed in collaboration with professional graphic artists. Feedback was gathered from the expert panel, and modifications were made based on their recommendations to ensure clarity and relevance. For example, adjustments were made to the posture illustrations to ensure they accurately reflected real-world scenarios.

Both the self-developed Traditional Chinese (Taiwanese) questionnaire, which underwent reliability and validity testing, and the English version of the questionnaire, which was translated through a two-stage Chinese-to-English and English-to-Chinese process, are provided in the Appendix A. The English version has not undergone reliability and validity testing.

### 2.2. Expert Content Validity Review

After the questionnaire was designed, an expert panel comprising five specialist physicians (mean years of professional experience: 17.8 ± 4.5 years), nine physical therapists (12.4 ± 8.0 years), and two occupational therapists (12.0 ± 7.1 years) were invited to evaluate the appropriateness of its textual content (1 = very inappropriate, 4 = very appropriate), the clarity of its descriptions (1 = very unclear, 4 = very clear), and the relevance of its illustrations (1 = very irrelevant, 4 = very relevant). These experts were highly experienced in managing patients with low back pain, with more than half of their patients presenting with such complaints. These experts reviewed each question, text, and illustration and provided feedback and suggestions. Their recommendations and suggestions constituted the basis for revising the text and illustrations.

### 2.3. Questionnaire Reliability

Participants were recruited from the Taipei City Hospital Yangming Branch Rehabilitation Department through direct invitation by medical staff. The inclusion criteria were patients aged 40 and above diagnosed with low back pain, capable of communicating in Mandarin or Taiwanese, and willing to complete the questionnaire. Exclusion criteria included illiteracy, central nervous system injuries (such as stroke or Parkinson’s disease), acute severe pain, or other structural issues.

A total of 100 participants were recruited for this study, which aimed to assess both the internal consistency and the test–retest reliability of the newly developed posture assessment questionnaire. The sample size of 100 was selected based on prior studies that investigated the reliability of test–retest and Cronbach’s alpha estimates. The previous studies recommended that sample sizes smaller than 100 were unreliable for such analyses, while samples of at least 100 participants yielded robust reliability estimates [21,22]. Cronbach’s α was calculated to evaluate the internal consistency of each questionnaire item and measure the strength of the correlation between items within each category. An α value of ≥0.7 was considered indicative of high internal consistency [23]. Test–retest reliability was assessed to determine the questionnaire’s stability over time [21]. A total of 100 participants completed the questionnaire in two rounds, with a 2-week interval between the first and second rounds. Intraclass correlation coefficients (ICCs) were calculated to determine the stability of responses over time, reflecting the consistency between the two test administrations.

### 2.4. Statistical Analysis

The expert content validity review was conducted online using Google Forms, and the item-level content validity index (I-CVI) was calculated. This index indicates the validity of individual items. For each item, the I-CVI is calculated on the basis of the number of experts who rated the item as either 3 or 4, effectively categorizing the scale into relevant and irrelevant [24]. In this study, the I-CVI was evaluated for textual content, description clarity, and illustration relevance. All statistical analyses including the calculation of Cronbach’s alpha, ICCs, and I-CVI were performed using SAS software (version 9.4; SAS Institute, Cary, NC, USA).

## 3. Results

The study included 100 participants (63 males and 37 females) with a mean age of 57.33 years (±7.21) (Table 1). Most participants were aged between 55 and 64 years (44.0%), and 62.0% were employed. Additionally, 87.0% were married, and common conditions reported included lumbar degeneration (38.0%), chronic low back pain (28.0%), and intervertebral disc herniation (26.0%). Table 2 presents the ergonomic and non-ergonomic postures for each of the six daily activities assessed in the questionnaire, along with their corresponding textual descriptions. These descriptions were designed to clearly distinguish between ergonomically safer postures and those that may increase the risk of back pain or injury. The questionnaire’s content validity was assessed through the expert review using the I-CVI, evaluating textual content, the descriptions’ clarity, and the illustrations’ relevance for each of the six posture categories (Table 3). The I-CVI values for both “ergonomic” and “non-ergonomic” postures were consistently high across all categories, with most scores exceeding 3.5 out of 4.0. The high I-CVI scores reflect strong content validity, confirming that the questionnaire’s items were both relevant and precise in conveying the postures being assessed. All of the illustrations included in the questionnaire are also available in the Appendix A.

For the first category (lifting heavy objects), the experts provided varying feedback on the textual content. For example, their suggestions included specifying whether the description involved lifting objects close to the body, emphasizing that the feet should be positioned for support during lifting, and describing how the hips and knees should bend while keeping the body upright during squatting. Regarding the clarity of the descriptions, the experts also had diverse opinions; for example, they recommended providing additional details on the exertion method (e.g., using the shoulders rather than the forearms and wrists) and noted that the description of squatting appeared to be odd. Despite these varying opinions on textual content and description clarity, all experts rated the illustrations as 4 (very relevant), indicating a unanimous agreement on the importance of visual representation despite variations in the textual descriptions (Figure 1).

For the second category (sitting), the experts provided several suggestions for improving the clarity of the descriptions and content relevance. They recommended including precise phrases to describe postures; for example, they recommended specifying how the lower back should lean against the back of the chair. They also noted that the description of the slouched sitting posture was insufficient and suggested further elaboration on what “average leaning against the back of the chair” entails. Regarding images, they recommended that images should be included to depict the shoulders and arms hanging naturally or hands placed on the knees, the chair’s back being vertical to the ground, and hands positioned naturally. Although they recommended further precision and elaboration in the textual descriptions, their illustration relevance scores remained high. These discussions simplified the textual descriptions. In addition, the illustrations were revised to reflect a vertical chair back, naturally hanging shoulders and arms, and depictions of computer postures and mobile phone use.

For the third category (putting on shoes), the experts provided varying suggestions. Some of them recommended specifying that standing while putting on shoes was less common and that individuals were more likely to squat or bend at the waist or sit on a chair and bend over. Others emphasized that in evaluations of postures that involve standing on one foot or on both feet while putting on shoes, the key point is whether the waist is inappropriately bent. Additionally, some experts suggested adding a description of placing the feet on the ground and directly bending over to put on shoes. In terms of sitting while putting on shoes, the experts recommended highlighting the importance of raising the foot and keeping the back straight. These suggestions served as the basis for the revision of the original text, and illustrations were added to help participants better understand that the purpose of this category was to differentiate between maintaining a straight back and bending over while putting on shoes.

For the fourth category (face washing and tooth brushing), two experts recommended that the text specify whether the body is upright and leaning forward or bent forward at the waist. They recommended focusing on whether the knees were bent or kept straight in conjunction with the body’s upright or forward-bending posture. One expert suggested discussing the squat motion, which should address the sequence of muscle and joint activation and core stability, because an inappropriate technique may lead to injury. Because this motion can be particularly challenging for certain groups, appropriate explanations regarding breath control should be provided.

For the fifth category (getting out of bed), the experts had no major comments on the textual descriptions or illustrations. However, they suggested including a description of whether one uses their hands to push themselves off from the bed; they recommended that the text specify whether one should use both hands or one hand to support the body against the bed.

For the sixth category (doing sit-ups), the experts primarily recommended modifying the illustrations to a side view because this view would provide a more precise depiction. Sit-ups are a familiar home exercise for strengthening core muscles. However, because they involve excessive flexion of the lumbar spine or incorrect postures, they can easily lead to an excessive compressive force on the lumbar spine, which may lead to lumbar spine and low back injuries.

The internal consistency of the questionnaire was evaluated using Cronbach’s α, and the results revealed an α value of 0.808, indicating high internal consistency across the items. Test–retest reliability was assessed using ICCs, with a subset of participants completing the questionnaire in two rounds separated by a 2-week interval. The ICC value was 0.78, indicating high reliability and stability over time.

## 4. Discussion

Overall, the internal consistency of the proposed questionnaire, as reflected by a Cronbach’s α of 0.808, suggests that its items were well-aligned and consistently measured the intended construct. This α value exceeded the commonly accepted threshold of 0.7 for high internal consistency [25], indicating that the questionnaire items were sufficiently interrelated without redundancy. This high consistency across items suggests that the respondents had similar interpretations of the questions and that the questionnaire effectively captured the core dimensions of postures related to low back pain [25]. In addition, the test–retest reliability of the proposed questionnaire, with an ICC of 0.78, indicates its stability over time. This ICC value, which was within the range of 0.75 to 0.9, indicates the questionnaire’s high reliability, confirming its robustness in repeated application [26]. This consistency across a 2-week interval suggests that the respondents’ responses remained stable over time, which is essential for clinical and research applications in which reliable repeated measures are required. In summary, the questionnaire’s high performance in the two aforementioned reliability metrics confirms its suitability in clinical and research settings for evaluating postural habits and identifying potential contributors to low back pain.

The visualized questionnaire offers several advantages over traditional text-based assessments; in particular, it enhances comprehension and accuracy [27,28]. Its illustrations provide an intuitive approach for respondents to interpret and respond to questions regarding their daily life postures. According to the literature, visuals improve comprehension in 98% of studies, enabling the respondents to better understand and process complex information [29,30]. Mayer’s cognitive theory of multimedia learning supports the notion that combining images with text activates dual channels in the brain, enhancing learning and retention [31].

In this study, the questionnaire’s illustrations likely minimized misinterpretations of the postural descriptions because the respondents could easily form connections between images and their own experiences, which may have led to more accurate self-reporting. The experts unanimously agreed on the relevance of these illustrations, indicating the importance of visuals in ensuring that the respondents fully grasped the intended meaning of each question. This approach is particularly essential for populations with relatively low literacy levels or cognitive difficulties, for whom text alone may present challenges [29]. Hence, visual cues can render questionnaires more accessible and easier to use, improving the overall reliability of the data collected.

Developing a visualized posture assessment questionnaire has major practical implications in clinical and rehabilitation settings. In traditional settings, text-based assessments of daily postures pose challenges, particularly for older adults and individuals with relatively low literacy levels [32]. Integrating images into questionnaires facilitates comprehension, which can thus minimize the need for extensive reading or interpretation. This approach can be particularly useful for patients with cognitive impairments or for non-native speakers [33]. It can also help clinicians and therapists be more confident in relying on the patients’ self-reported data because the visual format mitigates the risk of misinterpretation. Furthermore, this approach can effectively identify posture-related problems and provide guidance for appropriate interventions, thereby supporting more tailored and precise rehabilitation strategies.

Compared with widely used posture assessment tools such as the NMQ and RMDQ, the proposed visualized questionnaire is more accessible and easier to use [34]. Although the NMQ and RMDQ have been validated and extensively used in clinical research, they primarily rely on textual descriptions, which can be limiting for specific patient populations. Therefore, integrating images into the questionnaire provides an intuitive method for respondents to evaluate their postures. This study’s high I-CVI scores for textual descriptions and illustrations indicate that the visualized questionnaire effectively captures relevant postural information. This innovative approach can bridge the gap between traditional text-based assessments and the need for more accessible, user-friendly tools in healthcare.

Despite the positive findings regarding the questionnaire’s reliability and validity, this study has some limitations that should be addressed in future research. First, this study was conducted within a regional teaching hospital, which may limit the generalizability of the findings. Future research should aim to include a broader range of populations and settings to confirm the questionnaire’s reliability and applicability across diverse demographic groups. Second, while previous studies have shown that a 2-week interval is commonly used in health research, this study focused on short-term reliability and did not evaluate the test–retest reliability over an extended period. To better understand the questionnaire’s consistency over time, future studies should incorporate longitudinal analyses to assess its long-term reliability in tracking posture-related behaviors. Third, although the proposed visualized questionnaire improves comprehension, further research is warranted to determine its applicability across different age groups, cultural contexts, and literacy levels. Finally, this questionnaire was developed in Traditional Chinese. The English used in this article was translated through a two-stage Chinese-to-English and English-to-Chinese translation process. However, further validation and reliability testing of the English version of the questionnaire are needed. Future studies could consider cross-cultural adaptation and validation to enable its use in other linguistic and cultural settings. The images in the questionnaire are likely to be universally applicable, but if modifications are needed, please feel free to contact the authors. Exploring the integration of more advanced technologies, such as augmented reality, could also provide new opportunities for enhancing the precision and user engagement of posture assessments.

## 5. Conclusions

This study developed a visualized text-based questionnaire. Testing confirmed this questionnaire’s high reliability and validity, rendering it a robust tool for evaluating daily postures related to low back pain. Integrating images into the questionnaire improved comprehension, making the questionnaire more accessible to diverse populations. Overall, this questionnaire can be used in clinical and research settings to identify posture-related problems and guide targeted interventions. Nevertheless, further research with more diverse samples is warranted to explore the questionnaire’s long-term reliability and applicability across various demographic groups.

## Figures and Tables

**Figure 1 healthcare-12-02274-f001:**
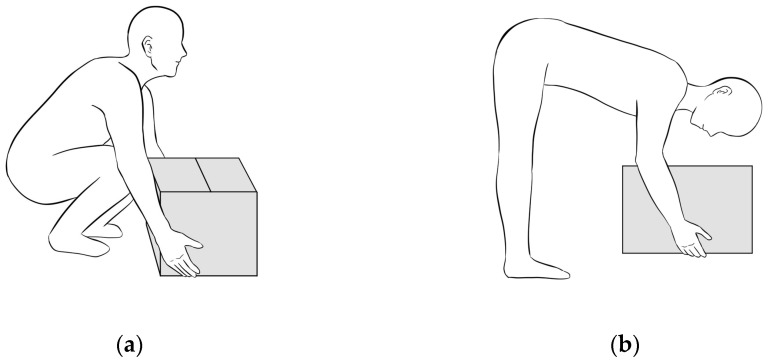
Ergonomic and non-ergonomic methods for lifting heavy objects. (**a**) The ergonomic posture for lifting heavy objects: lifting with knees bent and back straight while squatting. (**b**) The non-ergonomic posture for lifting heavy objects: lifting with knees straight and back bent while leaning forward.

**Table 1 healthcare-12-02274-t001:** Participant demographic data *.

Demographic Variable	Toral Participants (N = 100)	Male (N = 63)	Female (N = 37)
Age (mean ± S.D.)	57.33 ± 7.21	57.16 ± 6.42	57.62 ± 8.48
**Age Group**			
≤44 years	4 (4.0%)	1 (1.6%)	3 (8.1%)
45–54 years	33 (33.0%)	22 (34.9%)	11 (29.7%)
55–64 years	44 (44.0%)	31 (49.2%)	13 (35.1%)
≥65 years	19 (19.0%)	9 (14.3%)	10 (27.0%)
**Employment Status**			
Employed	62 (62.0%)	49 (77.8%)	13 (35.1%)
**Marital Status**			
Single	8 (8. 0%)	5 (7.9%)	3 (8.1%)
Married	87 (87.0%)	57 (90.5%)	30 (81.1%)
Divorced	1 (1.0%)	1 (1.6%)	0 (0.0%)
Widowed	4 (4.0%)	0 (0.0%)	4 (10.8%)
**Disease**			
Chronic low back pain	28 (28.0%)	19 (30.2%)	9 (24.3%)
Intervertebral disc herniation	26 (26.0%)	18 (28.6%)	8 (21.6%)
Lumbar degeneration	38 (38.0%)	26 (41.3%)	12 (32.4%)
Spondylolisthesis	8 (8.0%)	0 (0.0%)	8 (21.6%)

* Values are expressed as absolute numbers, with percentages in parentheses.

**Table 2 healthcare-12-02274-t002:** Ergonomic and non-ergonomic postures for daily activities and their posture descriptions.

Daily Activities and Postures	Postures	Posture Description
Lifting heavy objects	Ergonomic	Lifting with knees bent and back straight while squatting
	Non-ergonomic	Lifting with knees straight and back bent while leaning forward
Sitting	Ergonomic	Sitting upright with your back resting fully against the chair
	Non-ergonomic	Slouching posture with your back hunched and not resting fully against the chair
Putting on shoes	Ergonomic	Putting on shoes with knees bent and back straight
	Non-ergonomic	Putting on shoes while sitting, lifting your feet, and keeping your waist straight
Face washing and tooth brushing	Ergonomic	Washing face or brushing teeth with slightly bent knees and a straight back
	Non-ergonomic	Washing face or brushing teeth with knees straight and leaning forward with a bent back
Getting out of bed	Ergonomic	Rolling to your side first and pushing yourself up with your hand
	Non-ergonomic	Sitting up directly by bending at the waist
Doing sit-ups	Ergonomic	Performing a sit-up by lifting the entire back off the bed
	Non-ergonomic	Performing a sit-up by lifting only the shoulder blades off the bed

**Table 3 healthcare-12-02274-t003:** Item-level content validity index (I-CVI) for the six categories of daily activities and postures in the questionnaire.

Daily Activities and Postures	Postures	Textual Content	Descriptions’ Clarity	Illustrations’ Relevance
Lifting heavy objects	Ergonomic	3.77	3.38	4.00
	Non-ergonomic	3.69	3.85	4.00
Sitting	Ergonomic	3.92	4.00	4.00
	Non-ergonomic	3.92	3.54	3.85
Putting on shoes	Ergonomic	3.31	3.54	3.23
	Non-ergonomic	3.62	4.00	4.00
Face washing and tooth brushing	Ergonomic	3.85	3.92	3.92
	Non-ergonomic	4.00	4.00	4.00
Getting out of bed	Ergonomic	3.92	3.69	3.69
	Non-ergonomic	3.92	3.92	3.85
Doing sit-ups	Ergonomic	4.00	3.92	4.00
	Non-ergonomic	3.85	3.92	3.62

## Data Availability

Upon request from the corresponding author.

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
