# Peer review of "Development and Validation of a Visualized Posture Risk Assessment Questionnaire for Low Back Pain in Daily Activities: A Study in Taiwan"

_healthcare, 2024, doi:10.3390/healthcare12222274_

Round 1

Reviewer 1 Report

Comments and Suggestions for Authors

Dear Authors,

Thank you for submitting your manuscript on the development and validation of a visualized posture assessment questionnaire. Your work addresses an important area in evaluating posture-related low back pain, offering a potentially valuable tool that combines text and illustrations for improved accessibility and comprehension. However, in light of current perspectives in pain science and movement variability, as well as the broader application of the biopsychosocial model in pain management, several areas of your study warrant further consideration. Below, I provide detailed feedback and recommendations to strengthen both the conceptual framework and methodological rigor of your work.

In the context of current understandings in pain science and movement, the dichotomy of “correct” and “incorrect” postures warrants reconsideration. Numerous studies emphasize that movement and postural variability are natural and adaptive, rather than a source of dysfunction or pain when deviated from a so-called "ideal" posture. Persisting with terms like "correct" or "incorrect" posture may inadvertently foster hypervigilance and limit natural variability in movement, potentially reinforcing unhelpful beliefs about bodily vulnerability or mechanical causation of pain. I recommend that the authors revisit the terminology used to describe posture in the questionnaire and consider more neutral, context-sensitive alternatives such as “energy-efficient” or “less-strain” postures. This approach aligns with emerging research suggesting that a wide range of movement patterns can be adaptive and that focusing on variability and functional utility—rather than static correctness—may support a more nuanced understanding of pain and movement. Adjusting the language to reflect a spectrum of movement strategies, rather than a rigid dichotomy, could make the questionnaire more applicable and beneficial for diverse patient populations, especially in light of the biopsychosocial model now commonly embraced in pain management.

On the other hand, the introduction would benefit from a more explicit explanation of the specific gap this questionnaire addresses in relation to existing tools like the RMDQ or NMQ. Please clarify how the proposed questionnaire differs in scope and usability from these established instruments.

The methodological design is generally sound, but further details are needed: Given that the questionnaire was developed in Traditional Chinese, it would be useful to mention whether any cross-cultural adaptations or pilot testing were conducted, especially considering the potential for use in other linguistic or cultural settings.

Regarding statistical analysis, while the reported Cronbach’s alpha of 0.808 indicates good internal consistency, further discussion is warranted regarding whether this level of reliability varies across the different posture categories. Please provide Cronbach’s alpha values per category if available or explain why this was not done.

The discussion rightly emphasizes the questionnaire's potential utility, but the limitations should be more explicit. Specifically, the focus on a regional teaching hospital may limit the generalizability of the findings. Future research should include more diverse populations to verify long-term reliability and applicability across different demographic groups.

As your study focuses on short-term reliability, it would be helpful to highlight plans or recommendations for testing the questionnaire’s performance over extended periods in clinical settings.

In summary, your study presents a valuable and innovative contribution to the assessment of posture in low back pain. However, addressing the suggested revisions—particularly the reconsideration of terminology to align with contemporary movement science and enhancing methodological transparency—will be crucial for ensuring the tool’s relevance and applicability across diverse populations. I hope that these recommendations provide helpful direction, and I look forward to seeing how your revised manuscript will further strengthen this important area of research.

Author Response

Comments 1: In the context of current understandings in pain science and movement, the dichotomy of “correct” and “incorrect” postures warrants reconsideration. Numerous studies emphasize that movement and postural variability are natural and adaptive, rather than a source of dysfunction or pain when deviated from a so-called "ideal" posture. Persisting with terms like "correct" or "incorrect" posture may inadvertently foster hypervigilance and limit natural variability in movement, potentially reinforcing unhelpful beliefs about bodily vulnerability or mechanical causation of pain. I recommend that the authors revisit the terminology used to describe posture in the questionnaire and consider more neutral, context-sensitive alternatives such as “energy-efficient” or “less-strain” postures. This approach aligns with emerging research suggesting that a wide range of movement patterns can be adaptive and that focusing on variability and functional utility—rather than static correctness—may support a more nuanced understanding of pain and movement. Adjusting the language to reflect a spectrum of movement strategies, rather than a rigid dichotomy, could make the questionnaire more applicable and beneficial for diverse patient populations, especially in light of the biopsychosocial model now commonly embraced in pain management.
Response: Thank you for this insightful comment. We appreciate your perspective on the evolving understanding of movement and pain science, particularly in relation to postural variability and the biopsychosocial model. In response to your suggestion, we have revised the terminology in the questionnaire to avoid rigid labels like "correct" and "incorrect." Instead, we have adopted the terms "ergonomic" and "non-ergonomic" postures, which focus on postures that may minimize or increase physical strain, respectively. This adjustment aims to reflect a more adaptable and functional understanding of posture, aligning with contemporary pain science by avoiding dichotomous labels that could promote hypervigilance or reinforce beliefs about bodily vulnerability.
Modification: Throughout the manuscript and questionnaire, we have replaced "correct" and "incorrect" posture with "ergonomic" and "non-ergonomic" posture. This change supports a more nuanced approach to posture assessment, encouraging diverse movement strategies in line with the biopsychosocial model and making the questionnaire more applicable to a broader range of patient populations. For example, in lines 103-107: "Each category includes two indicators: an ergonomic posture and a non-ergonomic posture, with descriptions and illustrations for each. Ergonomic postures minimize strain on the musculoskeletal system, promoting safety and comfort. In contrast, non-ergonomic postures are associated with increased physical strain and a higher risk of discomfort or injury [20]."

Comments 2: On the other hand, the introduction would benefit from a more explicit explanation of the specific gap this questionnaire addresses in relation to existing tools like the RMDQ or NMQ. Please clarify how the proposed questionnaire differs in scope and usability from these established instruments.

Response: Thank you for this valuable feedback. We agree that clarifying the unique scope and usability of our questionnaire in comparison to existing tools such as the RMDQ and NMQ would enhance the introduction. In response, we have revised the Introduction to emphasize that while the RMDQ and NMQ are widely used to assess musculoskeletal health and disability, they primarily rely on text-based questions and do not specifically address posture-related behaviors in daily activities. Our questionnaire aims to fill this gap by incorporating visual illustrations and evaluating ergonomic and non-ergonomic postures, which enhances usability, especially for populations with limited literacy or visual impairments. This tool provides a more accessible, behavior-focused assessment approach for identifying postural habits related to low back pain.

Modification: In the Introduction section, we added an explanation of how the proposed questionnaire fills a specific gap by addressing posture-related behaviors and offering a visually accessible format, distinguishing it from traditional text-based tools like the RMDQ and NMQ; please refer to lines 70-81 for the revised text.

Comments 3: The methodological design is generally sound, but further details are needed: Given that the questionnaire was developed in Traditional Chinese, it would be useful to mention whether any cross-cultural adaptations or pilot testing were conducted, especially considering the potential for use in other linguistic or cultural settings.

Response: Thank you for your thoughtful suggestion regarding cross-cultural adaptation and pilot testing. We developed and validated this questionnaire specifically in Traditional Chinese for a Taiwanese population. At this stage, no formal cross-cultural adaptation or pilot testing has been conducted in other linguistic or cultural settings. However, we recognize the potential value of expanding this tool’s applicability to diverse populations. To address this, we have included a statement in the Discussion section, suggesting the need for future studies to conduct cross-cultural adaptations and validations should the questionnaire be used in non-Taiwanese populations.

Modification: In the Discussion section, we added the following statement (please refer to lines 329 for the revised text.): “However, further validation and reliability testing of the English version of the questionnaire are needed. Future studies could consider cross-cultural adaptation and validation to enable its use in other linguistic and cultural settings."

Comments 4: Regarding statistical analysis, while the reported Cronbach’s alpha of 0.808 indicates good internal consistency, further discussion is warranted regarding whether this level of reliability varies across the different posture categories. Please provide Cronbach’s alpha values per category if available or explain why this was not done.

Response: Thank you for your valuable suggestion on the statistical analysis. We understand the interest in examining Cronbach’s alpha across different posture categories. However, Cronbach’s alpha is generally calculated for the entire scale to assess overall internal consistency. Calculating alpha for each category individually, especially when each category has a limited number of items, can lead to lower reliability scores and may not accurately reflect the questionnaire’s internal consistency.

To address this, we performed an analysis where we removed each category item individually and recalculated Cronbach’s alpha, which yielded values ranging from 0.709 to 0.805. This analysis suggests that the internal consistency remains acceptable when each item is examined independently. Given these results, we reported the overall Cronbach’s alpha for the full scale to reflect the questionnaire’s reliability as a unified tool, following standard psychometric practices.

Comments 5: The discussion rightly emphasizes the questionnaire's potential utility, but the limitations should be more explicit. Specifically, the focus on a regional teaching hospital may limit the generalizability of the findings. Future research should include more diverse populations to verify long-term reliability and applicability across different demographic groups.

Response: Thank you for your valuable feedback regarding the limitations of our study. We agree that the focus on a regional teaching hospital may limit the generalizability of our findings. In response, we have revised the Discussion section to explicitly acknowledge this limitation and emphasize the need for future studies to include more diverse populations and settings. Expanding the participant pool across various demographic groups will be essential to verify the questionnaire’s long-term reliability and applicability beyond the current study population.

Modification: In the Discussion section, we added the following statement (please refer to lines 302-306 for the revised text.): “First, this study was conducted within a regional teaching hospital, which may limit the generalizability of the findings. Future research should aim to include a broader range of populations and settings to confirm the questionnaire’s reliability and applicability across diverse demographic groups.”

Comments 6: As your study focuses on short-term reliability, it would be helpful to highlight plans or recommendations for testing the questionnaire’s performance over extended periods in clinical settings.

Response: Thank you for your insightful suggestion. We agree that evaluating the questionnaire’s performance over extended periods would provide valuable insights into its long-term reliability in clinical settings. In response, we have added a statement in the Discussion section, recommending future studies to conduct longitudinal assessments of the questionnaire. This approach would help determine its consistency and applicability over time, especially for monitoring posture-related behaviors in relation to low back pain.

Modification: In the Discussion section, we added the following statement (please refer to lines 320- for the revised text.): “Second, while previous studies have shown that a two-week interval is commonly used in health research, this study focused on short-term reliability and did not evaluate the test–retest reliability over an extended period. To better understand the questionnaire’s consistency over time, future studies should incorporate longitudinal analyses to assess its long-term reliability in tracking posture-related behaviors.”

Comments 7: In summary, your study presents a valuable and innovative contribution to the assessment of posture in low back pain. However, addressing the suggested revisions—particularly the reconsideration of terminology to align with contemporary movement science and enhancing methodological transparency—will be crucial for ensuring the tool’s relevance and applicability across diverse populations. I hope that these recommendations provide helpful direction, and I look forward to seeing how your revised manuscript will further strengthen this important area of research.

Response: Thank you for your encouraging and constructive summary. We are grateful for your recognition of our study’s contribution to assessing posture in relation to low back pain. We have carefully considered and implemented your suggestions to refine the terminology in alignment with contemporary movement science and have made adjustments to enhance the methodological transparency of our work. We believe these revisions will improve the tool’s relevance and applicability across diverse populations. We appreciate your guidance, which has been invaluable in strengthening the manuscript, and we hope the revised version meets your expectations.

Reviewer 2 Report

Comments and Suggestions for Authors

Introduction

Authors should specify what is meant by correct posture.

The term poor posture is generalistic and not very specific.  The authors should refer to the risk factors that determine the appearance of pain from the postural point of view.

Line 54-56: The authors should give a more technical explanation of why the lumbar musculature is forced.

Line 62: Duplicate information. The authors already refer in the text to the socioeconomic impact of low back pain.

Line 66: Determine the choice of proposed scale examples and reference.

On line 71 the authors indicate that the proposed questionnaires are determinant for assessing low back pain. Two of them assess musculoskeletal pain and only one assesses low back disability. I suggest revising the wording of this paragraph of the introduction.

Materials and Methods

The sample recruitment process is not explained.

Line 129: The reference data of the statistical test must be referenced.

Results

The data shown in the table 1 are not identifiable in percentages, they are expressed in absolute numbers.

Line 161: The authors refer to the fact that the questionnaire has also been validated in English. This should be explained in the methods, not in the results.

Author Response

Comments 1: Authors should specify what is meant by correct posture. The term poor posture is generalistic and not very specific.  The authors should refer to the risk factors that determine the appearance of pain from the postural point of view.

Response: Thank you for your guidance. Consistent with the feedback from the first reviewer, we have reconsidered the use of terms like "correct" and "incorrect" to describe posture in the questionnaire. As noted in Comments 1, recent insights in pain science and movement emphasize that movement and postural variability are natural and adaptive, rather than inherently dysfunctional when they differ from an "ideal" posture. Using terms like "correct" and "incorrect" could unintentionally reinforce rigid beliefs about bodily vulnerability or pain causation and limit natural movement variability. To align with this perspective, we have adopted the terms "ergonomic" and "non-ergonomic" to describe postures that may either minimize or increase physical strain, respectively. This adjustment reflects an adaptable and functional understanding of posture, aligning with the biopsychosocial model commonly used in pain management.

Modification: Throughout the manuscript and questionnaire, we have replaced "correct" and "incorrect" with "ergonomic" and "non-ergonomic." For example, in lines 97-101, we now specify: "Each category includes two indicators: an ergonomic posture and a non-ergonomic posture, with descriptions and illustrations for each. Ergonomic postures minimize strain on the musculoskeletal system, promoting safety and comfort. In contrast, non-ergonomic postures are associated with increased physical strain and a higher risk of discomfort or injury [21]." This terminology change provides a more nuanced approach to posture assessment, allowing for diverse movement strategies that are in line with modern pain science and applicable to a broader range of patient populations.

Comments 2: Line 54-56: The authors should give a more technical explanation of why the lumbar musculature is forced.

Response: Thank you for this suggestion. We agree that a more technical explanation would enhance clarity. In response, we have revised Lines 54-56 to provide a detailed explanation of the biomechanical forces acting on the lumbar musculature. Specifically, we describe how certain postures, such as bending or twisting, can increase compressive and shear forces on the lumbar spine, thereby placing additional strain on the supporting musculature. This clarification aims to give readers a better understanding of the mechanisms through which improper posture may lead to lumbar strain and potential discomfort.

Modification: In Lines 53-57, we added a technical explanation detailing how specific movements increase compressive and shear forces on the lumbar spine, thus increasing the load on the lumbar musculature and contributing to potential musculoskeletal strain. "Inappropriate movements, such as excessive bending or twisting, can significantly increase compressive and shear forces on the lumbar spine. These forces strain the lumbar musculature, requiring the muscles to stabilize the spine under load. Over time, this increased demand on the lumbar muscles may lead to muscle fatigue, strain, and an elevated risk of low back pain."

Comments 3: Line 62: Duplicate information. The authors already refer in the text to the socioeconomic impact of low back pain.

Response: Thank you for pointing out the redundancy regarding the socioeconomic impact of low back pain. We have reviewed the text and removed the duplicate information in Line 62 to streamline the content and avoid repetition.

Comments 4: Line 66: Determine the choice of proposed scale examples and reference.

Response: Thank you for your valuable suggestion regarding the choice of scale examples and references. In response, we have clarified the selection of assessment tools in the revised manuscript. Specifically, we have referenced the Roland–Morris Disability Questionnaire (RMDQ) and Nordic Musculoskeletal Questionnaire (NMQ) as widely used tools for assessing musculoskeletal health due to their validated reliability. We also included the Rapid Upper Limb Assessment (RULA) as an example of a semi-visual tool primarily designed for occupational settings. However, we acknowledge that these tools primarily target work-related or task-specific postures and rely on text-based assessments, which may limit their accessibility for some populations. This limitation highlights the need for a daily life posture assessment tool that is visually accessible and applicable in a broader context.

Modification: In the manuscript, we revised the text to specify our choice of RMDQ, NMQ, and RULA as commonly used tools for musculoskeletal and posture assessment, while also addressing their limitations in assessing daily life activities. Please refer to Lines 68-77 for the updated text.

Comments 5: On line 71 the authors indicate that the proposed questionnaires are determinant for assessing low back pain. Two of them assess musculoskeletal pain and only one assesses low back disability. I suggest revising the wording of this paragraph of the introduction.

Response: Thank you for your suggestion to clarify the scope of the referenced tools in relation to low back pain. Consistent with Reviewer 1’s Comment 2, we have revised this paragraph in the Introduction to specify that the RMDQ and NMQ primarily assess general musculoskeletal pain, with the RMDQ additionally assessing low back disability. We have also clarified that these tools focus more on work-related or task-specific postures and may not directly address posture-related behaviors in daily life. This adjustment provides a clearer distinction among the tools referenced and contextualizes the need for our proposed questionnaire, which emphasizes daily posture assessment through a visual format.

Modification: In the Introduction section, we revised the paragraph to specify that RMDQ and NMQ are general musculoskeletal assessment tools, with RMDQ addressing low back disability specifically, and to clarify their limitations in assessing daily life posture behaviors. Please refer to Lines 66–79 for the updated text.

Comments 6: Materials and Methods: The sample recruitment process is not explained.

Response: Thank you for your feedback regarding the Materials and Methods section. We agree that clarifying the sample recruitment process and referencing the statistical test data will enhance the transparency of our study. In response, we have updated the Materials and Methods section to include a detailed explanation of the sample recruitment process, outlining the inclusion and exclusion criteria, as well as the recruitment sources. 

Modification: In the Materials and Methods section, we have clarified the sample recruitment process. Please refer to Lines 133-139 for these updates.

Comments 7:  Line 129: The reference data of the statistical test must be referenced.

Response: Thank you for your feedback regarding the Materials and Methods section. we have added the appropriate reference for the statistical test used in Line 129 to ensure clarity and reproducibility.

Comments 8: Results: The data shown in the table 1 are not identifiable in percentages, they are expressed in absolute numbers.

Response: Thank you for your helpful comment regarding the presentation of data in Table 1. We have revised the table to ensure that both absolute numbers and percentages are clearly displayed. Now, each value is presented as an absolute number followed by the corresponding percentage in parentheses (e.g., 4 (4.0%)). Additionally, we have added a footnote to the table to clarify that values are expressed as absolute numbers with percentages in parentheses. We hope this adjustment improves the clarity and readability of the data.

Modification: Please refer to the updated Table 1, where absolute numbers and percentages are clearly indicated.

Comments 9: Line 161: The authors refer to the fact that the questionnaire has also been validated in English. This should be explained in the methods, not in the results. 

Response: Thank you for this valuable comment. We agree that the information regarding the translation and validation of the English version of the questionnaire would be more appropriately placed in the Materials and Methods section. We have moved this information accordingly, while retaining only the relevant findings in the Results section.

Reviewer 3 Report

Comments and Suggestions for Authors

I would like to thank the authors for submitting the scientific article.

1. Reviewer

With the anti-plagiarism programme Turnit in, a match of 27% was found; after omitting citations, bibliography and less than 1% match, the match is finnaly 6%.

2. Reviewer:

The title of the article "Development and Validation of a Visualized Posture Assessment Questionnaire for Low Back Pain: A Study in Taiwan" suggests to the reader that it is a visual questionnaire on low back pain. In reality, however, it is a risk assessment of posture for low back pain-related postures (see Line 33 and Line 81 in the yellow highlighted appendix) during daily activities

Kindly I ask the authors to change the title of the article in line with the idea in Line 33. It should be clear from the title what type of questionnaire it is. Does it assess low back pain or the risk of developing low back pain.

Thank you

3. Reviewer:

ABSTRACT

The summary contains all the elements that make the theme clear. With in the range of about 250 words.

Kindly  I ask the authors to replace part of the sentence "...daily actions..." in Lines 22 and 23 (yellow in the appendix) "with daily activities or similar.

Thank you.

4. Reviewer:

KEY WORDS

Keywords in the correct order.

Kindly I would like to ask the authors for a correction, namely that this questionnaire is about risk assessment for low back pain. Ad key word about risk assessment.

Thank you.

5. Reviewer:

INTRODUCTION

The introduction to the problem is clearly described, as is the reason for creating a visual questionnaire.

I politely ask the authors to correct "….....to wear…...." (Line 56) It literally means that they are already dressed, the action is completed, please correct. Also corecction in questionarie and Table 2.

Thank you.

6. Reviewer

MATERIALS AND METHODS

The materials and methods are clearly formulated. They are described in sufficient detail to allow others to replicate and build on published results.

The statistical processing of the data is appropriate.

Thank you.

7. Reviewer

RESULTS

The results for this type of analysis are presented in a simple, clear and understandable way.

Please authors. To think about Figure 1. the correct position for lifting a load from the knees. In this knee position when lifting a load, the right knee, in this example, bears the maximum load. In a correct squat, both knees must bear the load when lifting. This image is not displayed correctly. Please correct the image. The text states that both knees are bent during a squat. In the picture, this is a squat with support on one foot. In a real squat, you are leaning on both feet. Perhaps you should emphasise in the text that both feet provide support.

Thank you.

8. Reviewer

DISCUSSION

The authors present the data obtained in a simple way and emphasise the reliability of the questionnaire. They emphasise the advantages of this questionnaire compared to standardised questionnaires in text form and the importance of simple questionnaires for the needs of illiterate, visually impaired people and people with cognitive difficulties. They named the limits of this questionnaire

I would ask the authors to proofread the article again to improve the quality of the English language.

Thank you.

9. Reviewer

CONCLUSIONS

The conclusion is reasonable and consistent with the obtained data and the relevant publications used in writing the research article.

Thank you.

 10. Reviewer

REFERENCES

Written according to the journal gudielines.

Thank you.

The authors present a simple questionnaire that can be used to analyse the risk and relationship between poor posture in daily activities and lower back pain. This type of questionnaire can provide an insight into the postural habits of the respondents as it can be easily completed by the respondents. The simplicity of the application also allows illiterate people to circle the appropriate answer with a verbal explanation. People with various disabilities, low vision or cognitive difficulties can also give an answer about their postural habits, although we must be careful here as there are significant limitations within the degree of cognitive impairment. Furthermore, the same questionnaire can be completed by children if we rely only on the visual representation of items. Simplicity like the Visual Analogue Scale, which is widely used in the initial assessment of pain in the general population. A valuable attempt that needs to be further considered and elaborated as stated in limitations.

Author Response

Comments 1: With the anti-plagiarism programme Turnit in, a match of 27% was found; after omitting citations, bibliography and less than 1% match, the match is finnaly 6%.

Response: Thank you for conducting the plagiarism check using Turnitin. We appreciate your thorough review. We acknowledge that the initial match rate was 27%, but as noted, this was largely due to citations, bibliography, and minor matches. After omitting these, the final match rate is 6%, which is within acceptable limits for original content. We are committed to upholding academic integrity, and we appreciate your efforts in verifying the originality of our work.

Comments 2: The title of the article "Development and Validation of a Visualized Posture Assessment Questionnaire for Low Back Pain: A Study in Taiwan" suggests to the reader that it is a visual questionnaire on low back pain. In reality, however, it is a risk assessment of posture for low back pain-related postures (see Line 33 and Line 81 in the yellow highlighted appendix) during daily activities

Kindly I ask the authors to change the title of the article in line with the idea in Line 33. It should be clear from the title what type of questionnaire it is. Does it assess low back pain or the risk of developing low back pain.

Response: Thank you for your insightful feedback regarding the title. We agree that a more precise title would clarify the focus of the questionnaire on assessing posture-related risks associated with low back pain in daily activities. In response, we have revised the title to better align with the study's content.

Revised Title:
"Development and Validation of a Visualized Posture Risk Assessment Questionnaire for Low Back Pain in Daily Activities: A Study in Taiwan"

Comments 3: Kindly  I ask the authors to replace part of the sentence "...daily actions..." in Lines 22 and 23 (yellow in the appendix) "with daily activities or similar.

Response: Thank you for your suggestion to improve the wording in the abstract. We have replaced the phrase "daily actions" with "daily activities" to enhance clarity and consistency throughout the text.

Modification in Abstract:
"The proposed questionnaire evaluates six categories of daily activities, including lifting heavy objects, sitting, wearing shoes, face washing and tooth brushing, getting out of bed, and doing sit-ups, or similar actions."

Comments 4: KEY WORDS. Keywords in the correct order. Kindly I would like to ask the authors for a correction, namely that this questionnaire is about risk assessment for low back pain. Ad key word about risk assessment.

Response: Thank you for your suggestion to improve the keywords. We have added "risk assessment" as a keyword to better represent the focus of the questionnaire on evaluating the risk of low back pain-related postures.

Revised Keywords:
low back pain; posture; daily activities; risk assessment; questionnaire validation; visualized questionnaire; Taiwan

Comments 5: The introduction to the problem is clearly described, as is the reason for creating a visual questionnaire. I politely ask the authors to correct "….....to wear…...." (Line 56) It literally means that they are already dressed, the action is completed, please correct. Also corecction in questionarie and Table 2.

Response: Thank you for your attention to these details. We appreciate your observation, as our questionnaire emphasizes the posture involved in the action of putting on shoes. In response, we have replaced "to wear" with "putting on" throughout the manuscript to better highlight the posture during this action.

Modification: All instances of "to wear" have been changed to "putting on" to emphasize the posture associated with the act of putting on shoes.

Comments 6: The materials and methods are clearly formulated. They are described in sufficient detail to allow others to replicate and build on published results. The statistical processing of the data is appropriate.

Response: Thank you for your positive feedback on the Materials and Methods section and the statistical processing. We appreciate your acknowledgment of the clarity and detail provided, as well as the appropriateness of the statistical methods used. We hope these elements contribute to the replicability and robustness of our study.

Comments 7:  The results for this type of analysis are presented in a simple, clear and understandable way. Please authors. To think about Figure 1. the correct position for lifting a load from the knees. In this knee position when lifting a load, the right knee, in this example, bears the maximum load. In a correct squat, both knees must bear the load when lifting. This image is not displayed correctly. Please correct the image. The text states that both knees are bent during a squat. In the picture, this is a squat with support on one foot. In a real squat, you are leaning on both feet. Perhaps you should emphasise in the text that both feet provide support.

Response: Thank you for your valuable feedback regarding Figure 1. We appreciate your insight on the correct positioning for lifting a load with both knees supporting the weight. We have revised the image to accurately display a proper squat position, ensuring that both knees are shown to bear the load evenly.

Comments 8: The authors present the data obtained in a simple way and emphasise the reliability of the questionnaire. They emphasise the advantages of this questionnaire compared to standardised questionnaires in text form and the importance of simple questionnaires for the needs of illiterate, visually impaired people and people with cognitive difficulties. They named the limits of this questionnaire. I would ask the authors to proofread the article again to improve the quality of the English language.

Response: Thank you for your positive feedback on our discussion and for highlighting the importance of our questionnaire's accessibility for diverse populations. We appreciate your suggestion to improve the quality of the English language. We have carefully proofread the article to ensure clarity and precision in the language used.  Additionally, this manuscript was edited by Wallace Academic Editing to ensure the accuracy and quality of the English language.

Comments 9: The conclusion is reasonable and consistent with the obtained data and the relevant publications used in writing the research article.

Response: Thank you for your positive feedback on the conclusion. We are glad to hear that you found it reasonable and consistent with our data and the relevant literature. We appreciate your review and valuable comments.

Comments 10: REFERENCES. Written according to the journal gudielines.

Response: Thank you for your positive feedback on the references section. We have ensured that all references are formatted according to the journal guidelines. We appreciate your thorough review.

Comments 11: The authors present a simple questionnaire that can be used to analyse the risk and relationship between poor posture in daily activities and lower back pain. This type of questionnaire can provide an insight into the postural habits of the respondents as it can be easily completed by the respondents. The simplicity of the application also allows illiterate people to circle the appropriate answer with a verbal explanation. People with various disabilities, low vision or cognitive difficulties can also give an answer about their postural habits, although we must be careful here as there are significant limitations within the degree of cognitive impairment. Furthermore, the same questionnaire can be completed by children if we rely only on the visual representation of items. Simplicity like the Visual Analogue Scale, which is widely used in the initial assessment of pain in the general population. A valuable attempt that needs to be further considered and elaborated as stated in limitations.

Response: Thank you for your positive feedback on the simplicity and accessibility of our questionnaire. We appreciate your recognition of its potential to assess postural habits in diverse populations, including individuals with low vision, cognitive difficulties, or limited literacy. We acknowledge the limitations regarding the degree of cognitive impairment, and we agree that further exploration and refinement are necessary to maximize the tool's applicability across various demographic groups. Your insights are valuable, and we will consider these aspects in future studies to improve the questionnaire’s usability and adaptability.

Round 2

Reviewer 1 Report

Comments and Suggestions for Authors

Thanks for your detailed revisions and for giving such thoughtful attention to the feedback offered in the original review. I especially appreciate the changes, including changing "correct" and "incorrect" to "ergonomic" and "non-ergonomic," which meets some of the important shifts in contemporary understandings of movement and pain science and gives such a much greater sense of inclusion with the questionnaire.

Your addition to the introduction seems to make the tool so different from pre-existing tools, like the RMDQ and NMQ, by underlining the added value of visual aids in monitoring daily posture-related behaviors. The extension of the section on limitations by suggesting a future cross-cultural adaptation and assessment of the longitudinal reliability greatly enhances transparency and applicability.

In addition, the rationale for presenting an overall Cronbach's alpha and further item-by-item analysis strengthens the internal consistency of the questionnaire in supporting the unified reliability score. The discussed revisions, along with your openness toward future validations in diverse settings, strengthen the clinical relevance of this innovative tool.

Again, thank you for your thorough work.